# Smartphone-supported Positive Adjustment Coping Intervention (PACI) for couples undergoing fertility treatment: A randomised controlled trial

Franziska Kremer[1], Maren Schick[1], Sabine Roesner[2], Ariane Germeyer[2], Markus Moessner[3], Stephanie Bauer[3], Beate Ditzen[1,4], Tewes Wischmann[1,4]*

**1** Institute of Medical Psychology, University Hospital Heidelberg, Center for Psychosocial Medicine, Heidelberg, Germany, **2** Department of Gynecological Endocrinology and Fertility Disorders, University Women's Hospital Heidelberg, Heidelberg, Germany, **3** Center for Psychotherapy Research, University Hospital Heidelberg, Center for Psychosocial Medicine, Heidelberg, Germany, **4** Heidelberg University, Heidelberg, Germany

* tewes.wischmann@uni-heidelberg.de

## Abstract

### Background

Couples undergoing fertility treatment experience emotional stress. Meta-analyses show heterogeneous results regarding the effectiveness of psychosocial interventions on anxiety and depression.

### Objective

The aim of this study was to reduce psychosocial stress (anxiety, depression) in infertile patients. In addition, pregnancy rates and perceived distraction as well as perceived subjective relief were determined.

### Methods

In a two-arm RCT we used "Positive Adjustment Coping Intervention (PACI)". Participants received the first of 13 daily text messages on their smartphones on the day on which the cryopreserved embryos were thawed or one day after oocyte aspiration. Patients in the PACI condition obtained daily positive adjustment statements and were asked to relate these to their personal situation at least twice a day. Patients in the comparison condition were given daily think tasks providing cognitive distraction. Anxiety and depression scores were assessed with ScreenIVF and perceived distraction as well as relief with an evaluation form.

### Results

PACI did not reduce participant's depressive scores measured with standardized questionnaire ($P < .007$, N = 227), did not significantly change participant's anxiety

which permits unrestricted use, distribution, and reproduction in any medium, provided the original author and source are credited.

**Data availability statement:** All relevant data are within the paper and its Supporting Information files.

**Funding:** The author(s) received no specific funding for this work.

**Competing interests:** The authors have declared that no competing interests exist.

**Abbreviations:** ANOVA, Analysis of variance; ART, Assisted reproductive technology; CD, cognitive distraction; ICSI, Intracytoplasmic sperm injection; IVF, In-vitro fertilization; IVM, In-vitro-maturation; PACI, Positive Adjustment Coping Intervention; PRCI, Positive Reappraisal Coping Intervention; RCT, Randomized controlled trial; SD, Standard deviation; TESE, Testicular sperm extraction.

(N = 227), nor did it increase pregnancy rates (N = 191). PACI had a significant effect on subjective perceived distraction (*P* = .005, N = 197) and on perceived relief (*P* = .026, N = 197).

## Conclusion

This is the first RCT to use modern media to test a simple positive adjustment cognition intervention in women and men who underwent fertility treatment. The low-dose psychosocial intervention apparently was not sufficient to reduce emotional distress during the waiting period between embryo transfer and pregnancy test. Preferably, psychosocial support for infertile individuals could be provided face-to-face to reduce the target variables depression and anxiety. However, PACI has a positive influence on perceived distraction and perceived relief.

## Trial registration

ClinicalTrials.gov: NCT03118219 (July 9, 2019)

## Introduction

According to the WHO infertility is a "disease characterized by the failure to establish a clinical pregnancy after 12 months of regular, unprotected sexual intercourse or due to an impairment of a person's capacity to reproduce either as an individual or with his/her partner. " [1] (p. 1795). Approximately one in six people, or 17.5% of the world's adult population, suffer from infertility. Childlessness is often perceived as a life crisis imposing an emotional burden equivalent to that of a traumatic psychological event [2,3]. According to ESHRE guidelines, patients are no more depressed than the general population or comparable control groups before starting IVF treatment. There is conflicting evidence that patients are more anxious (state and trait anxiety) than the general population before starting their first IVF cycle [4]. Assisted reproductive technology (ART) is the method of choice for couples who still want to have children. The treatment involves considerable emotional stress [5]. In particular, the 14-day waiting period between embryo transfer and the subsequent pregnancy test is usually experienced by couples as emotionally burdensome [6,7]. This waiting period is often characterised both by increased depression and anxiety levels and more frequent somatic symptoms [6]. The development of anxiety resulting from the uncertainty and the sense of futility typically induced by waiting times in medical contexts [8]. Waiting is an integral part of the reproduction cycle, which contributes to infertility and its treatment being low-control chronic stressors [9]. At this stage, the sense of control is greatly mitigated, as little can be done to facilitate the implantation of the fertilised egg in the uterus and improve the prospects of pregnancy.

When patients are informed that ART- treatment has been unsuccessful, 1–2 out of 10 women show clinically significant depressive symptoms. After receiving the

pregnancy test for their IVF/ICSI treatment, one in 4 women and 1 in 10 men suffer from a depressive disorder. One in 7 women and one in 20 men suffer from an anxiety disorder [4].

Face-to-face psychosocial interventions have a significant impact on depression in infertile women, but according to a recent meta-analysis, they do not significantly improve anxiety scores. A trend does however appear to be emerging in pregnancy rates [10].

Psychosocial infertility counselling is not always available locally. If it is provided, patients frequently do not avail themselves of it for fear of stigmatisation [11]. For this reason, technology-based interventions have advantages in connection with the provision of low-threshold support services: no additional clinic appointments are required, financial costs are lower [12,13].

With a view to providing support for patients during the waiting period invariably involved in infertility treatment, Lancastle and Boivin [7] developed a low-threshold psychological intervention called "Positive Reappraisal Coping Intervention" (PRCI). PRCI consists of an introductory text and a paper card with 10 core statements embodying positive thoughts and/ or behaviours and is based on cognitive reappraisal techniques. Initial feasibility studies [7,14] have demonstrated the effectiveness and acceptability of this intervention for women coping with the waiting period between embryo transfer and pregnancy test while undergoing ART.

For this study, we used the approved German version of PRCI and modified it to "Positive Adjustment Coping Intervention (PACI)". Instead of "reappraisal" we used the German term "Neuausrichtung", translated here as "adjustment". A person´s ability to adapt effectively to the demands of the environment and to the stress caused by those demands can be expressed in terms of psychological adjustment [15]. Psychological adjustment to difficult life crises is multifactorial and includes adequate coping strategies, sufficient social support and the absence of negative cognitions. Quality of life depends on how successfully one adapts to life's challenges. Inadequate adjustment is associated with anxiety or depression [16].

Apart from PRCI, to the best of our knowledge, there is only one other self-administered coping intervention, designed specifically for patients faced with medical waiting periods at home: cognitive distraction [17]. Women and men undergoing genetic cancer-risk assessment indicate that these distraction interventions (e.g., "count to 50 and visualise the numbers in your head"; "take some exercise—keeps you fit and takes your mind off your worries") are helpful in reducing the pressure on participants with high baseline stress [17–19].

The primary outcome of the study is the reduction of psychosocial distress (and therefore increasing the quality of life) during waiting period involved in infertility treatment. The effectiveness of PACI in comparison with cognitive distractions during the waiting period, was tested with a pre-post assessment using ScreenIVF [20]. Observation focused on the subscales depression and anxiety. Our hypothesis was that PACI would be more effective in reducing psychosocial distress than cognitive distraction(CD). The secondary outcome is the impact of the intervention on women's pregnancy rates. Another secondary outcome addressed in this study is the participants´ subjective evaluation of the effectiveness of the intervention. PACI is expected to provide distraction and relief during the waiting phase, a period typically characterised by rumination and worry.

## Methods

We applied the CONSORT reporting guidelines [21] and created a CONSORT checklist (S1 File). The protocol of the present study was published in advance [22] and the study was registered on the Clinical Trials website (www.clinicaltrials. gov, trial number NCT03118219). We did not make any significant changes to our methods after trial commencement. The study was approved by the Ethics Committee of Heidelberg University Faculty of Medicine (S-074/2017).

## Study design

The effectiveness of a smartphone-assisted psychosocial intervention for women and men undergoing reproductive treatment was investigated in a two-arm RCT. During the waiting phase after in-vitro fertilization (IVF) or intracytoplasmic

sperm injection (ICSI) treatment, participants were randomly assigned to either the PACI condition or the comparison condition (cognitive distraction). To assess the impact of PACI (Table 1 and Fig 1), participants in both conditions were asked to complete questionnaires at three time points: 1.) immediately before the waiting period (pre-intervention = T0), 2.) on day 13 of the 14-day waiting period (post-intervention = T2) and 3.) one month after the waiting period (evaluation = T3).

## Participants and recruitment

The participants were enrolled from the fertility clinic at the University Women´s Hospital, Heidelberg, Germany, between August 1, 2017 and June 30, 2019. Participants who terminated treatment prematurely have already been analysed [23]. The study population consists of couples undergoing fertility treatment using fresh or cryopreserved embryos with IVF or ICSI including in-vitro-maturation (IVM) and testicular sperm extraction (TESE). Before thawing the cryopreserved fertilised eggs or after egg puncture, the doctors informed all couples about the study. A total of 401 people were invited to participate in the study, of which 308 expressed interest to participate.

Inclusion criteria were willingness to participate in the study, written participation consent, own smartphone (capable of receiving text messages and displaying internet links), and disclosure of the relevant mobile phone number. As all study materials (including PACI) were only available in German, women and men were excluded from the study if their command of the German language was inadequate. If one partner of a couple refused to participate, the other partner could still participate in the study as a single participant. Participants could withdraw their consent to participate at any time without giving reasons and without disadvantages for the continuation of the medical care administered to them or their partners. All in all, of 93 participants were excluded for various reasons (Fig 1).

At this point, 308 patients received the first questionnaire, which was designed to assemble socio-demographic information (age, education, occupation, marital status, children, duration of partnership) and gynaecological and reproductive facts (duration of desire to have a child, duration of fertility treatment, cause of fertility treatment). The subjects were also given the ScreenIVF.

In the case of fertilisation failure, degeneration of all cryopreserved embryos after thawing, or medical reasons for the termination of embryo transfer (embryonic arrest, ovarian hyperstimulation syndrome, etc.), all text messages were

**Table 1. Positive Adjustment Coping Intervention: schedule of enrolment, interventions and assessment.**

|  | Study period | | | | |
| --- | --- | --- | --- | --- | --- |
|  | Enrolment | Allocation | Post-allocation | | Follow-Up |
| Timepoint | T0 |  | T1 | T2 | T3 |
|  |  |  | Waiting period |  |  |
| Enrolment: |  |  |  |  |  |
| Eligibility screen | x |  |  |  |  |
| Informed consent | x |  |  |  |  |
| Mobile phone number | x |  |  |  |  |
| Allocation |  | x |  |  |  |
| Interventions: |  |  |  |  |  |
| PACI Condition |  |  | ←———————→ |  |  |
| CD Condition |  |  | ←———————→ |  |  |
| Assessments: |  |  |  |  |  |
| Sociodemographics | x |  |  |  |  |
| SCREEN-IVF | x |  |  | x |  |
| Pregnancy |  |  |  | x | x |
| Evaluation |  |  |  |  | x |

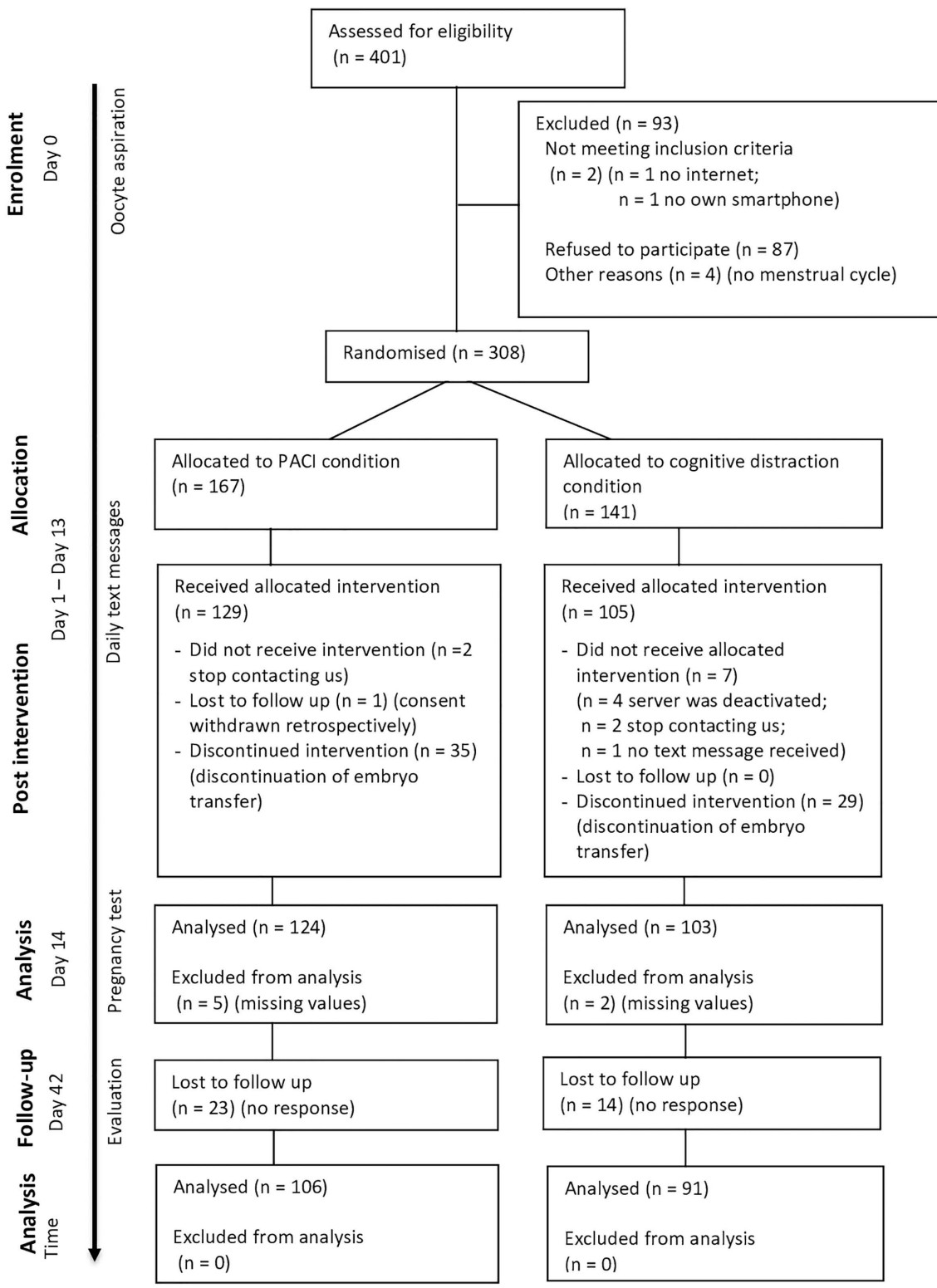

**Fig 1. The study flow chart.** The detailed inclusion and exclusion process and the number of patients at each stage.

discontinued as soon as the study team had been informed. As the text messages are restricted to the waiting time between oocyte puncture/thawing and embryo transfer, participants with cancelled embryo transfers were excluded from the study (n = 64). We also excluded participants who stopped responding (n = 4), who did not receive the text messages (n = 1) or whose online evaluation platform was temporarily unavailable (n = 4).

Finally, 124 participants in the PACI condition and 103 in the cognitive distraction condition received intervention and were included in the post intervention analysis (Fig 1). In the final analysis, 106 participants were included in the PACI condition and 91 participants in the distraction condition.

### Sample size

For sample size calculation, a mean effect size of d = 0.25 and a correlation of r = 0.5 between repeated measures were assumed (quoted in 30). The calculation of the effect size and the power analysis were carried out in advance with the statistics programme G*Power [24]. The RCT was designed as a two-arm study. Therefore, a total of 158 participants needed to be recruited (79 participants per condition) to test the condition differences with a power of 95% at an alpha level of α = 0.05. Taking into account an estimated attrition rate of 30% [7,14], at least 55 participants therefore had to be recruited for each condition.

### Randomisation

Participants were randomised to one of the two intervention conditions by a statistician (T.W.) using a computerised randomisation system (spreadsheet). This took place on the day of oocyte puncture/thawing, i.e., on day 0 of the waiting period after the first assessment (T0: pre-intervention questionnaire). Randomisation was performed by a statistician who had no contact with participants at any time. The clinic staff were blinded and did not know which intervention the couples had been assigned to. Assignment of the woman and the man of a couple to the same intervention condition is designed to clearly distinguish the effects of each intervention. Patients were informed of their study condition assignment when they received the first text message (so there were not blinded).

### Intervention conditions

All patients willing to participate received an information leaflet about the study. After declaring their informed consent via signature, patients received an information booklet informing them about the two different interventions. Participants were then given brief written instructions pertaining to positive adjustment or cognitive distraction. Both conditions received the first of 13 daily text messages on their smartphones either on the day on which the cryopreserved embryos were thawed of one day after oocyte aspiration. Patients in the PACI condition received positive adjustment statements on a daily basis ("I'm going to do something good for myself today", "I want to be relaxed about the waiting situation", etc.). These were designed to encourage participants to take a more positive attitude to the waiting period. Women and men were advised to read the text message as soon as they received it and subsequently whenever they felt the need to do so, but were asked to relate the positive adjustment statement to their personal situation at least twice a day. Patients in the comparison condition were given daily think tasks (brainteasers) as a cognitive distraction ("The day before yesterday, Martin was still 35—next year he will be 38. When is his birthday? "; "What's the next number in this series: 1–2 - 3–5 - 8–13 - 21–34 -?", etc. with the correct solutions written backwards).

### Materials and study measures

At three specific time points, participants in the two conditions were asked to complete questionnaires. The data of the patients were collected via self-report questionnaires on paper (T0) and online (T1, T2) and also from the medical records. Due to the comprehensive recording of psychosocial distress during fertility treatment, we chose the German version of ScreenIVF [19] as a follow-up instrument to compare two different measurement time-points (T0, T1).

The ScreenIVF is validated for women and men [25,26]. The main measure of outcome are the pre-post differences in the ScreenIVF variables anxiety and depression. The secondary outcomes are the pregnancy rate one month after the intervention plus the participants´ subjective evaluation of the perceived effectiveness of PACI (T2). Sociographic variables, medical data and possible pregnancies at the time of measurement after the intervention were assessed as potential moderators.

To assess the clinical characteristics of the study participants, a pre-intervention questionnaire assembled socio-demographic information (age, education, occupation, marital status, children, duration of partnership) and gynaecological and reproductive facts (duration of childbearing, duration of fertility treatment, cause of fertility treatment). As a primary measure of outcome, all participants were asked to complete the 34-item ScreenIVF: state anxiety (10 items based on the short version of Spielberger State and Trait Anxiety Inventory [27]), depression (7 items of the short Beck Depression Inventory version [28]), social support (5 items) and, with regard to the perception of fertility problems, helplessness (6 items) and lack of acceptance (6 items). Each item was scored on a four-point Likert scale. The score for each scale was calculated by summing the responses for each item and considering them against clinically relevant cut-off scores (anxiety ≥24, depression ≥4, social support ≤15, helplessness ≥14, acceptance ≤11). The internal consistency of all ScreenIVF scales in our study is satisfactory: Cronbach's alpha for anxiety = 0.88, for depression = 0.76, for social support = 0.88, for helplessness = 0.81 and for acceptance = 0.87.

One day before the pregnancy test at the clinic, participants received a link via their smartphone and were asked to complete the post-intervention questionnaire on Unipark, an online research survey website (post-intervention). The questionnaire included the ScreenIVF and questions about whether the patients had previously taken a pregnancy test at home and if so, what the result was.

One month after the end of the waiting period (T2), participants received a link on their smartphones enabling them to access an evaluation questionnaire. The evaluation questionnaire was based on the 24-item intervention evaluation form proposed by Lancastle and Boivin [6] but consisted of selected six items only. Using a rating scale, participants were asked about acceptance (effectiveness and duration of effects), perception of psychological effects (distraction), endorsement (recommendation) and effects on stress during the waiting period (perceived relief). With an open ended question we also offered the participants the opportunity to make comments on the study. Furthermore, the participants were asked about their pregnancy status.

## Statistical methods

Missing data had been replaced. If 80% of the values per subscale were available from Screen IVF, the remaining 20% were replaced by the respective mean value of the subscale. Imputation of missing values using the respective mean value for ScreenIVF was only necessary in 13 cases for STAI and in one case for BDI, representing a total of 6.2% (pre-intervention).

The analysis was conducted blind to the condition. Descriptive statistics and t-tests plus $\chi^2$-test were used to check for differences in the baseline data between the two conditions. Continuous variables were expressed as mean with standard deviation (SD), while categorical variables were expressed as a number with percentage. With repeated measures, differences between condition group in the ScreenIVF variables were tested with a mixed factorial analysis of variance (ANOVA). The effects of different moderators were explored. For a sub-group analysis, pre-intervention anxiety scores were split into two groups using the median.

At the beginning of the calculations, we checked the preliminary assumptions for ANOVA. The Shapiro-Wilk tests and visual checks were performed to evaluate the normality of the distribution for continuous variables. Where appropriate, data were transformed. In order to identify outliers, we generated boxplots. Extreme outlier values scoring more than 3 times the interquartile range were removed from the analysis. The Levene test was used to test the homogeneity of the error variances between groups and the Box test to assess homogeneity of the covariance matrices. Since the power of

the Box test depends on the number of cases and normally distributed data, this test becomes significant more quickly the more cases there are. For this reason, some authors recommend changing the significance level to.025 [29,30].

If the preliminary assumptions for ANOVA were not given, we calculated the non-parametric Wilcoxon test for paired samples. Visual inspection of the histogram of difference scores was performed to evaluate the symmetrical distribution of scores.

The secondary outcome was the pregnancy rate of the sample according to intention to treat. For binary outcomes we calculated odds ratios, each with a 95% CI. The consensus group (2014) recommended live birth (defined as any delivery of a live child ≥20 weeks' gestation) or cumulative live birth as the preferred primary outcome for infertility trials. Our data collection period was 42 days. Accordingly, no live birth could be recorded.

The results of the evaluation survey were determined using the χ²-test for association. The precondition for performing this test is that all expected cell frequencies be greater than 5. This precondition was verified. All analyses were conducted using IBM SPSS Statistics V.27 [31].

## Results

### Characteristics of sample

The general characteristics of the sample are presented in Table 2 and 3. A total of 234 subjects participated, 148 female and 86 male. The average age of the women was 35.77 (SD = 4.267) that of the men 38.02 (SD = 5.778).

The average duration of fertility treatment was 2.3 years (SD = 1.849). The groups differed significantly in two variables. Both intervention groups differed in marital status with $P = 0.048$. Furthermore, causes of infertility differed significantly between the two intervention groups. In the PACI condition, the combined factor was reported less frequently as the cause of infertility than in the comparison condition. and groups were also not equally distributed. The cause for the difference was identified by applying the Bonferroni correction in the diagnosis subgroup "combined". In the PACI condition the number of the "combined" factor was lower as statistically expected, while the opposite was the case in the control condition. The baseline between the two intervention conditions was comparable in the other characteristics. Participants tended to have higher educational qualifications than the general population. As Table 2 shows, questions related to smoking and alcohol were left unanswered by the majority of the respondents.

### Depression

The ANOVA results for the depressive symptoms were not calculated because the data were strongly right-skewed despite data transformation (logarithmization). We calculated the non-parametric Wilcoxon test for paired samples. Visual inspection of the histogram of difference scores showed a symmetrical distribution of values. Participants' depressive symptoms increased in both conditions, $z = -2.681$, $P < .007$. The median of pre and post measurement was 1 in each case and the difference between the pre and post measurement was 0. The interquartile range was 2 in each case. The intervention did not improve depressive symptoms.

Furthermore, it should however be noted that the mean scores of depressive symptoms for both conditions did not exceed the cut-off ≥ 4 value [20] at either of the measurement times. Women and men did not display clinically relevant scores for these values (Table 4). However, it should be noted that at the first measurement point, 15.2% of participants reported a value higher than four, and at the second measurement point, 20.3% reported a value higher than four.

### Anxiety

Descriptive statistics (Table 5) show that at both measurement time points mean anxiety scores did not attain the clinically relevant cut-off value ≥ 24 [20]. Before the interventions, 21.4% of participants achieved a clinically relevant cut-off value. After the intervention, this Fig rose to 30.2%.

**Table 2. Overview of sociodemographic information of the sample.**

| Characteristic | PACI condition (n=129) | | CD condition (n=105) | | t/χ²-value statistic | P-value |
|---|---|---|---|---|---|---|
| | Women | Men | Women | Men | | |
| | (n=86) | (n=43) | (n=62) | (n=43) | | |
| Socio-demographic data | | | | | | |
| Age in years (M, ±SD) | 35.90 (4.04) | 38.95 (5.82) | 35.59 (4.60) | 37.12 (5,66) | 1.059 | .291 |
| Marital status (n, %) | | | | | 3.925 | .048 |
| Married | 69 (80.2) | 34 (79.1) | 56 (90.3) | 38 (88.4) | | |
| Living in partnership | 15 (17.4) | 9 (20.9) | 5 (8.1) | 5 (11.6) | | |
| Unanswered | 2 (2.4) | | 1 (1.6) | | | |
| Education (n, %) | | | | | 3.105 | .875 |
| No graduation | 1 (1.2) | 0 | 0 | 0 | | |
| Secondary school | 21 (25.3) | 7 (16.3) | 16 (26.2) | 13 (30.2) | | |
| Abitur | 10 (12.0) | 7 (16.3) | 6 (9.8) | 10 (23.3) | | |
| University | 44 (53.0) | 25 (58.1) | 35 (57.4) | 17 (39.5) | | |
| Postgraduate | 7 (8.4) | 4 (9.3) | 4 (6.6) | 3 (6.9) | | |
| Unanswered | | | 1 (1.6) | | | |
| BMI (n, %) | | | | | 3.955 | .556 |
| Mildly thin | 4 (4.8) | 0 | 1 (1.6) | 0 | | |
| Normal range | 62 (73.8) | 3 (17.6) | 37 (59,7) | 7 16.3) | | |
| Pre-obese | 9 (10.7) | 9 (52.9) | 14 (22.6) | 6 (14.0) | | |
| Obese (Class I) | 9 (10.7) | 2 (11.8) | 5 (8.1) | 2 (4.7) | | |
| Obese (Class II) | 0 | 3 (17.6) | 2 (3.2) | 2 (4.7) | | |
| Obese (Class III) | 0 | 0 | 0 | 2 (4.7) | | |
| Unanswered | 2 (2.4) | 26 (60.5) | 3 (4.8) | 24 (55.8) | | |
| Smoking (n, %) | | | | | .153 | .696 |
| Yes | 7 (8.1) | 2 (4.7) | 4 (6.5) | 2 (4.7) | | |
| No | 69 (80.2) | 23 (53.5) | 53 (85.5) | 23 (53.5) | | |
| Unanswered | 10 (11.6) | 18 (41.9) | 3 (4.8) | 17 (39.5) | | |
| Alcohol (n, %) | | | | | 1.229 | .746 |
| Never | 22 (25.6) | 8 (18.6) | 17 (27.4) | 7 (16.3) | | |
| Rarely | 19 (22.1) | 4 (9.3) | 13 (21.0) | 2 (4.7) | | |
| Now and then | 4 (4.7) | 3 (11.6) | 5 (8.1) | 3 (7.0) | | |
| Occasionally | 12 (14.0) | 5 (11.6) | 9 (14.5) | 8 (18.6) | | |
| Regularly | 0 | 0 | 0 | 0 | | |
| Unanswered | 29 (33.7) | 23 (53.5) | 16 (25.8) | 22 (51.2) | | |

Furthermore, pregnancies at the post-intervention time of measurement were assessed as potential moderators. Self-reported pregnancy test results, which participants had performed on their own initiative before post measurement, were available from 132 patients. The homogeneity of the error variances was determined by the Levene test ($P > .05$). The Box test ($P = .010$) indicated no homogeneity of covariances. A statistically significant interaction between anxiety and pregnancy was found, $F(1.00, 228.00) = 11.564$, $P = .001$, partial $\eta^2 = .08$. Participants' anxiety scores reduced when the pregnancy test was positive. Moreover, there was a significant interaction effect between anxiety, condition and pregnancy, $F(1.00, 228.00) = 5.579$, $P = .020$, partial $\eta^2 = .04$ (Figs 3 and 4).

**Table 3. Overview of the fertility information of the sample.**

| Characteristic | PACI condition (n = 129) | | CD condition (n = 105) | |
|---|---|---|---|---|
| | Women | Men | Women | Men |
| | (n = 86) | (n = 43) | (n = 62) | (n = 43) |
| Fertility data | | | | |
| Duration of desire to have children in years (M, ±SD) | 4.66 (3.23) | 4.22 (3.44) | 4.33 (2.59) | 4.90 (3.61) |
| Duration of infertility treatment in years (M, ±SD) | 2.42 (2.01) | 2.37 (2.12) | 2.22 (1.59) | 2.23 (1.61) |
| Cause of infertility (n, %) | | | | |
| Female -factor | 30 (34.9) | 13 (30.2) | 18 (29.0) | 8 (18.6) |
| Male -factor | 34 (39.5) | 15 (34.9) | 17 (27.4) | 13 (30.2) |
| Combined | 12 (14.0) | 10 (23.3) | 20 (32.3) | 17 (39.5) |
| Unexplained | 10 (11.6) | 5 (11.6) | 4 (6.5) | 4 (9.3) |
| Treatment type (n, %) | | | | |
| IVF | 33 (38.4) | 15 (34.9) | 20 (32.3) | 13 (30.2) |
| IVF, Cryopreservation | 0 | 0 | 1 (1.6) | 1 (2.3) |
| ICSI | 37 (43.0) | 24 (55.8) | 31 (50.0) | 23 (53.5) |
| ICSI, Cryopreservation | 6 (7.0) | 2 (4.7) | 4 (6.5) | 3 (7.0) |
| IVM | 5 (5.8) | 1 (2.3) | 4 (6.5) | 2 (4.7) |
| ICSI, TESE (testicular sperm extraction) | 5 (5.8) | 1 (2.3) | 0 | 0 |
| No. of births before study entry (n, %) | | | | |
| None | 63 (73.3) | 34 (79.1) | 46 (74.2) | 36 (83,7) |
| One | 19 (22.1) | 7 (16.3) | 12 (19.4) | 5 (11.6) |
| Two | 4 (4.7) | 2 (4.7) | 2 (3.2) | 1 (2.3) |
| No. of miscarriages (n, %) | | | | |
| None | 60 (69.8) | 30 (69.8) | 43 (69.4) | 33 (76,7) |
| One | 17 (19.8) | 7 (16.3) | 8 (12.9) | 5 (11.6) |
| Two | 4 (4.7) | 2 (4.7) | 3 (4.8) | 1 (2.3) |
| Three | 2 (2.3) | 1 (2.3) | 4 (6.5) | 2 (4.7) |
| Four | 2 (2.3) | 2 (4.7) | 1 (1.6) | 1 (2.3) |
| Five | 1 (1.2) | 0 | 1 (1.6) | 0 |
| Six | 0 | 1 (2.3) | 0 | 0 |

**Table 4. Descriptive statistics for depressive symptoms.**

| | | PACI condition | | | CD condition | | |
|---|---|---|---|---|---|---|---|
| | | Women | Men | Total | Women | Men | Total |
| Pre (T0) | M | 1.72 | 0.79 | 1.40 | 2.03 | 1.40 | 1.77 |
| | (±SD) | (2.12) | (1.01) | (1.86) | (2.19) | (1.73) | (2.03) |
| Post (T1) | M | 2.72 | 1.05 | 2.17 | 2.36 | 1.40 | 1.96 |
| | (±SD) | (3.34) | (1.74) | (3.01) | (3.03) | (2.36) | (2.80) |

Participants in the PACI condition experienced a decrease in anxiety scores when the pregnancy test was positive. Participants in the comparison condition remained at a similar level at both measurement time points. A negative pregnancy test resulted in a significant increase in anxiety scores in the PACI condition.

As a second factor, gender was included in another general linear model with repeated measures. There was no statistically significant interaction between time, gender and the study condition, $F(1.00, 223.00) = 0.007$, $P = .933$, partial $\eta^2 = .00$.

**Table 5. Descriptive statistics for state anxiety.**

|  |  | PACI condition | | | CD condition | | |
|---|---|---|---|---|---|---|---|
|  |  | Women | Men | Total | Women | Men | Total |
| Pre (T0) | M | 20.72 | 17.48 | 19.62 | 19.62 | 17.95 | 18.92 |
|  | (±SD) | (5.30) | (5.26) | (5.49) | (5.88) | (4.20) | (5.29) |
| Post (T1) | M | 21.65 | 18.12 | 20.45 | 20.62 | 18.56 | 19.76 |
|  | (±SD) | (5.56) | (5.54) | (5.78) | (6.56) | (5.24) | (6.10) |

There is no statistically significant interaction between time (T0, T1) and the study condition group, $F_{(1.00, 225.00)} = 0.00005$, $P = .995$, partial $\eta^2 = .000$ (Fig 2).

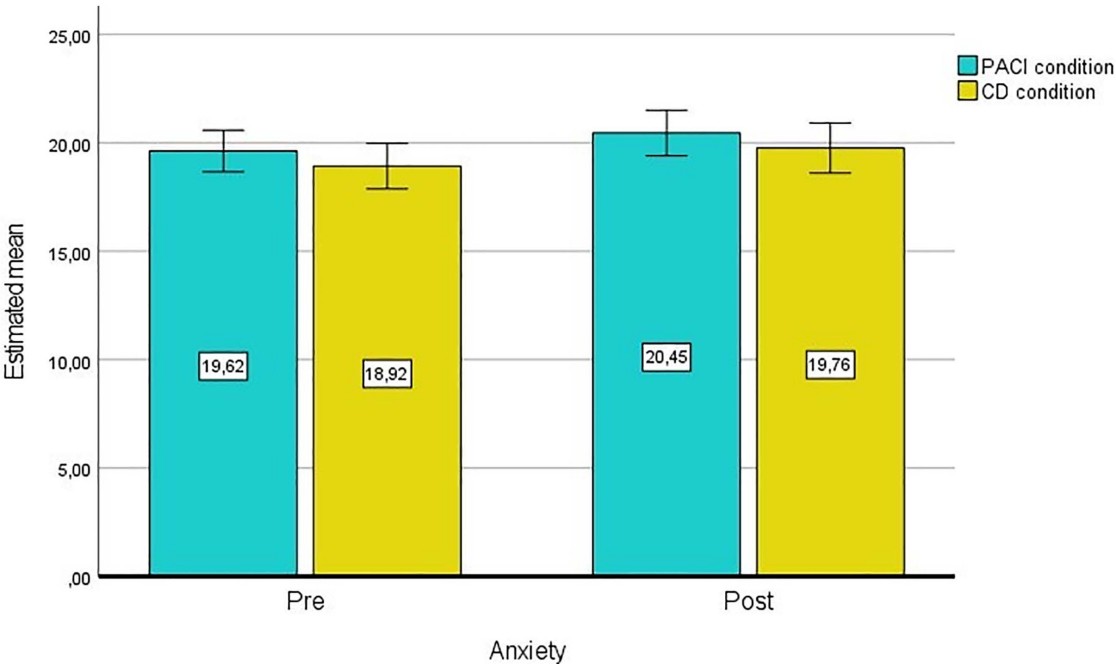

**Fig 2. General linear model (GLM) for anxiety with repeated measures and estimated means.**

Whether the duration of the desire to have children (in years) has an influence was also examined as covariate. There was a statistically significant interaction between time, duration of desire to have children and the study condition, $F_{(1.00, 222.00)} = 4.786$, $P = .03$, partial $\eta^2 = .02$ (Fig 5).

## Pregnancy rates

The secondary outcome was pregnancy rates one month after the intervention (follow-up, T2) as self-reported by the participants. With an Odds Ratio (OR) of 1.143 and a 95% Confidence interval (CI) 0.684 to 1.909 ($P = 0.611$), PACI did not increase pregnancy rates (Tables 6 and 7).

## Perceived distraction

Another secondary outcome was the perception of psychological effects, distraction, self-reported one month after the intervention ("Did the text messages provide a welcome distraction for you?").

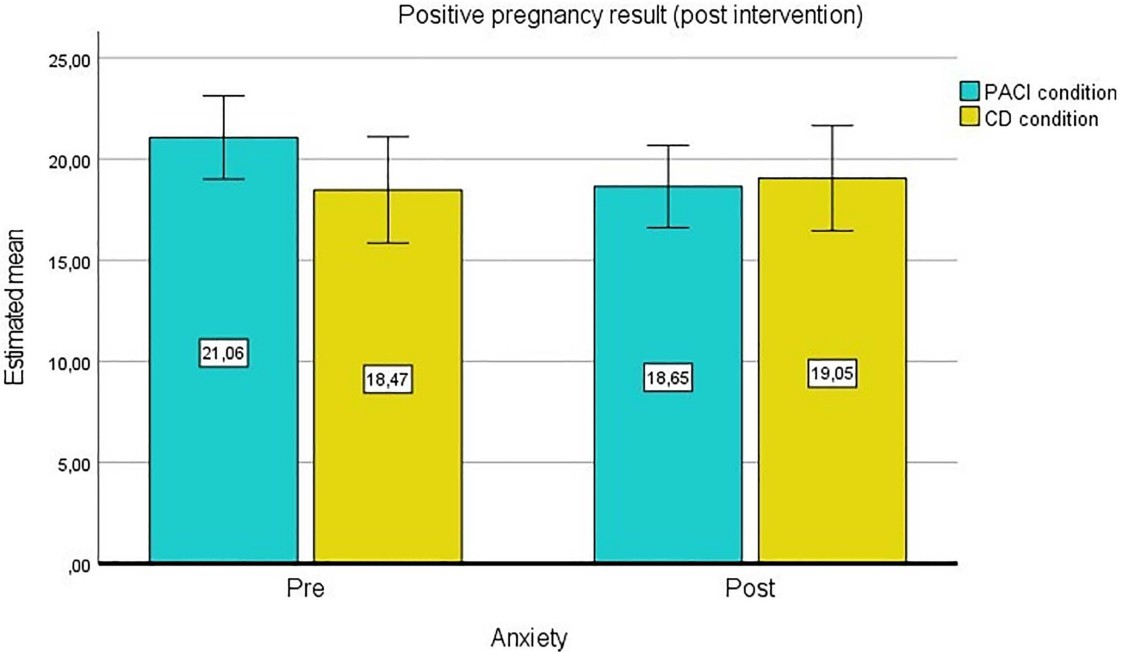

**Fig 3. GLM with pregnancy results as second factor, estimated means for anxiety (positive pregnancy result).**

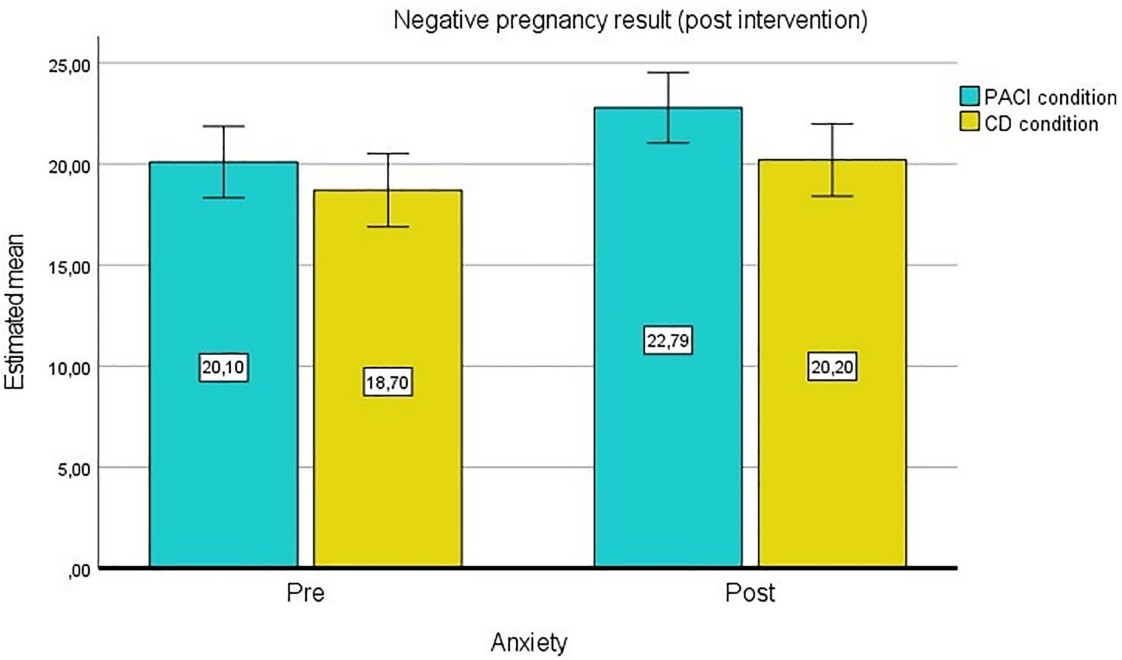

**Fig 4. GLM with pregnancy results as second factor, estimated means for anxiety (negative pregnancy result).**

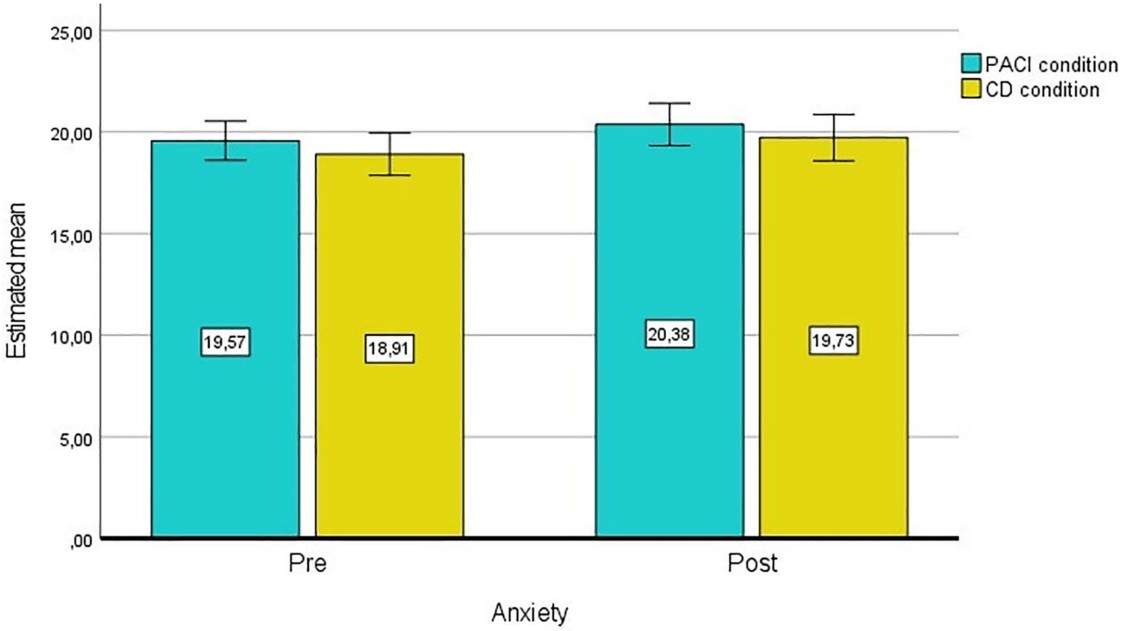

**Fig 5. GLM with covariate duration of desire to have children and estimated means for anxiety.**

**Table 6. Descriptive statistics for pregnancy with the sample size based on valid cases.**

|  | PACI condition (n = 103) | CD condition (n = 88) |
|---|---|---|
| Pregnant | 39 | 36 |
| Not pregnant | 64 | 52 |

**Table 7. Odds ratio for pregnancy with the sample size based on intention to treat.**

| Statistics | Values |
|---|---|
| Odds ratio | 1.1426 |
| 95% CI | 0.68 to 1.91 |
| Z-statistic | 0.509 |
| P-value | 0.611 |
| PACI condition ($n_{ITT}$ = 167) | Pregnant $n_{ITT}$ = 39<br>Not pregnant $n_{ITT}$ = 128 |
| CD conditio ($n_{ITT}$ = 141) | Pregnant $n_{ITT}$ = 36<br>Not pregnant $n_{ITT}$ = 105 |

A $\chi^2$-test for association was conducted between condition and distraction and all expected cell frequencies were greater than 5 (precondition). Results show a significant effect between condition and distraction, $\chi^2(4) = 14.94$, $P = .005$, $V = 0.275$. According to Cohen [32], the Cramer's V effect size measure can be interpreted similarly to a correlation. The post-hoc test with the Bonferroni correction showed that the significant $\chi^2$ was due to much more frequent agreement (text messages distract) in the PACI condition and less frequent agreement in the CD condition than expected (Table 8).

**Table 8. The cross table for perceived distraction one month after the waiting period.**

| | | Did the text messages provide a welcome distraction for you? | | | | | Total |
|---|---|---|---|---|---|---|---|
| | | Yes | Rather Yes | Neutral | Rather No | Don´t know | |
| PACI | Number | 37 | 22 | 20 | 16 | 11 | 106 |
| | % of PACI or CD group | 34.90% | 20.80% | 18.90% | 15.10% | 10.40% | 100.00% |
| | Corrected residues | 3.7 | −0.2 | −1.9 | −1.2 | −0.6 | |
| CD | Number | 11 | 20 | 28 | 20 | 12 | 91 |
| | % of PACI or CD group | 12.10% | 22.00% | 30.80% | 22.00% | 13.20% | 100.00% |
| | Corrected residues | −3.7 | 0.2 | 1.9 | 1.2 | 0.6 | |
| Total | Number | 48 | 42 | 48 | 36 | 23 | 197 |
| | % of PACI or CD^A group | 24.40% | 21.30% | 24.40% | 18.30% | 11.70% | 100.00% |

As a secondary outcome, the perceived duration of distraction by the text messages one month after the intervention was also collected ("Did this effect of the text messages last for a while?"). The $\chi^2$-test revealed a significant effect between condition and perceived duration of distraction, $\chi^2(3) = 15.54$, $P=.001$, $V=0.281$.

## Perceived relief

As an additional secondary outcome, we surveyed the participants' perceived relief one month after the intervention (T2). The $\chi^2$-test revealed a significant effect between condition and perceived relief, $\chi^2(4) = 11.09$, $P=.026$, $V=0.237$. The post-hoc test with the Bonferroni correction showed that the significant $\chi^2$ was due to much more frequent agreement (perceived relief) than expected in the PACI condition and less frequent agreement in the CD condition (Table 9).

Furthermore, the participants' recommendation and the time they spent were surveyed one month after the intervention. The $\chi^2$-test for patient recommendation showed a significant effect between condition and recommendation ($\chi^2(5) = 16.12$, $P=.007$, $V=0.286$), but not for condition and time spent ($\chi^2(3) = 3.18$, $P=.364$, $V=0.127$).

## Discussion

In the present study, no direct association was found between the subscales of ScreenIVF and the PACI intervention. Our result is consistent with a randomized controlled trial that examined the application of PRCI [14]. PRCI did not lead to a significant reduction in anxiety or depression or daily negative emotions during the waiting period. However, the participants reported significantly more positive emotions during the waiting period than the participants in the control group. They also reported that the application was easy to use and had a positive psychological effect. In summary, an

**Table 9. The cross table for perceived relief one month after the waiting period.**

| | | Did the text messages you received provide some relief during the waiting period? | | | | | Total |
|---|---|---|---|---|---|---|---|
| | | Yes | Rather Yes | Neutral | Rather no | No/ Don´t know | |
| PACI | Number | 22 | 17 | 21 | 19 | 27 | 106 |
| | % of PACI or CD group | 20.80% | 16.00% | 19.80% | 17.90% | 25.50% | 100.00% |
| | Corrected residues | 2.8378 | 1.0273 | −1.607 | −1.2558 | −0.3176 | |
| CD | Number | 6 | 10 | 27 | 23 | 25 | 91 |
| | % of PACI or CD group | 6.60% | 11.00% | 29.70% | 25.30% | 27.50% | 100.00% |
| | Corrected residues | −2.8378 | −1.0273 | 1.607 | 1.2558 | 0.3176 | |
| | Number | 28 | 27 | 48 | 42 | 52 | 197 |
| | % of PACI or CD group | 14.20% | 13.70% | 24.40% | 21.30% | 26.40% | 100.00% |

easy-to-administer low-dose intervention such as PACI does not appear to result in a clinically relevant reduction in the target variables of anxiety and depression in individuals undergoing fertility treatment. The effects of other psychosocial interventions (and other settings) on these target variables in this patient group are also unclear and contradictory, as a recent review with meta-analysis was able to show (109). Tailor-made interventions that specifically address the self-efficacy of individuals in fertility treatment could help to positively influence these target variables [33]. This should be examined in future studies.

We have found an indirect association: The self-performed pregnancy test result (before the collection of the anxiety scores) meditated the anxiety levels. A significant interaction effect was found between anxiety and pregnancy test result as well as between anxiety, pregnancy test result and condition. Once participants received a positive pregnancy test response, the anxiety levels decreased. The reverse was also observed. Another finding is that the desire to have children expressed in years had a significant effect as a covariate in the interaction between anxiety, condition and desire to have children expressed in years.

Regarding the ScreenIVF, patients were defined as being at risk when their scores on one of the five risk factors showed clinically relevant values (Verhaak et al. 2010). Our results did not show a clinically relevant risk profile, but this does not mean that the waiting time after embryo transfer and before the pregnancy test is not emotionally stressful. The (reported) prevalence rates for depression and anxiety increased over time. Our low-dose psychosocial intervention (in the form of positive adjustment sent as text messages) was apparently unable to bring about any relevant improvement in anxiety for the participants. In fact, the scores on the different scales tended to increase, which may possibly even indicate a worsening of psychological well-being. The waiting time between embryo transfer and pregnancy test is emotionally burdensome for the participants, and this burden obviously seems to increase over time. Patients have to be informed at an early stage about this concomitant feature of ART.

As Boivin and Lancastle [6] have shown, the uncertainty involved in waiting for the pregnancy test result was characterised by increasing anxiety and depression as the day of the pregnancy test approached. Lazarus and Folkman [34] claim that the immediacy of an important event is one of the crucial dimensions that may make people rate situations as stressful, and Boivin and Lancastle's findings support this claim. It should be noted, however, that in their study, negative emotions did not occur in the complete absence of positive emotions. The emotional profile during the waiting period was rather a combination of anxiety and positive emotions. Leiblum et al. [35] refer to this as "cautious optimism".

In our study, we used this type of cognitive distraction in a less specific form. Participants in the comparison group were daily given simple one-time brainteasers instead of repeatable interventions. Furthermore, since distractive focusing is not thought to be maladaptive in terms of well-being [36], we chose cognitive distraction in our study as a comparative intervention mildly distracting patients during the waiting period. This allows to assess whether receiving a text message is a distraction in itself.

Our study used this simple form of psychological positive adjustment. No significant effect on anxiety and pregnancy rates was achieved by our intervention, thus reaffirming the fact that psychological adjustment to difficult life crises is multifactorial, encompassing adequate coping strategies, sufficient social support and the absence of negative cognitions [16]. Further research is needed to verify whether more intensive positive adjustment cognition interventions would be effective.

We also found significant associations between perceived distraction and perceived relief in the two conditions. This implies that the participants in our PACI condition were able to benefit from the text messages because they felt them to be welcome distractions. This perceived effect even lasted for a while. Our results show that text messages provided patients with subjectively perceived relief. The participants benefiting from these effects indicated that they would recommend this low-dose intervention to other infertile individuals undergoing ART. The results of the present study suggest that participants in the PACI condition considered the test messages to be an acceptable and feasible intervention alleviating the experience of waiting for an IVF pregnancy test.

PACIs bring about subjectively perceived relief and distraction but do not have a significant clinically relevant effect on depression and anxiety. In order to shed more light on this aspect, it will be recommendable in future to collect the psychological variables, e.g., by using the momentary assessment/ daily diary method. Collected at various measurement points during the day, these variables may possibly show more specific effects of low-dose intervention.

Another aspect may be, that infertile individuals undergoing ART are more likely to ruminate and worry in the evening because during the day they are distracted by the daily grind. In the evening, the distraction provided by the working day or daily life in general is no longer present. The couples will return home and talk about their desire to have children. Therefore, it might be possible that an intervention in the evening might be more effective, this however needs to be evaluated further.

## Strengths and limitations

A number of other strengths and limitations of our study need to be discussed. The strengths of this randomized controlled trial include the large sample size and the fact that in our study of infertile individuals we addressed male partners involved.

Following up on previous research with regard to the PRCI [7,14,37–41], this study adds significantly to the scope of the existing data in two ways. First, we use modern media (smartphones) instead of a paper card with a list of statements. Most people often have their smartphone to hand [42]. Second, we assess both women and men, as on the psychological plane men have also proved to be affected by ART [43,44].

Participants in our sample had a higher than average level of education. This is likely to limit the generalisation of the results and the representativeness of our study probably suffers accordingly. In our study, self-assessment was also used (no objective measures). Furthermore, couples may have completed the questionnaire together and at the same time (contrary to instructions) or one partner may have persuaded the other to participate (study bias).

Our study compared two different types of intervention, positive adjustment coping and cognitive distraction. The absence of a control group is a weakness. Because no control group without any intervention was utilized, real clinical value cannot be exactly defined. For this reason, we have nothing to say about the way in which the scores on the subscales of the ScreenIVF would have developed without any intervention.

As a further limitation, a bias due to the participation of the respective partner is possible. People who participated without their partner may have felt less supported and their anxiety and depression levels may differ from those who participated with their partner.

Another weakness of our trial is that it is not possible to verify whether and how often the participants of both interventions groups read the text messages. If compatible with data-protection criteria, automated confirmation by technical services would be conceivable if text messages were opened and remained open for a certain time. This could function as a method of determining whether the text messages are read or not.

The evaluation of the interventions by the participants was carried out with the aid of a self-designed questionnaire. Distraction and relief were itemised and not measured via a standardized questionnaire. These potential limitations must be borne in mind when interpreting these results of the study.

## Conclusions

Taken together, the smartphone adapted version of the PRCI, here presented to infertile individuals in fertility treatment as positive reappraisal during the waiting period between embryo transfer and pregnancy test, did not reduce depressive symptoms, anxiety or increase pregnancy rates. However, the intervention was perceived as helpful to distract from the burden associated with the treatment and might have increased perceived control. This data is in line with research suggesting that individual daily stress levels or low mood by themselves usually do not impair fertility [45] and that a low-level "mini-intervention" alone therefore might not increase pregnancy rates. At the same time these results can contribute

to more tailor-made face-to-face interventions that address both partners at the same time and, thereby, help couples to successfully navigate in this challenging time during ART.

## Supporting information

**S1 File. Consort checklist.** Reporting checklist for randomised trial. Based on the CONSORT guidelines.
(PDF)

## Acknowledgments

We are grateful to all participants who agreed to take part in this study.

## Author contributions

**Conceptualization:** Maren Schick, Sabine Roesner, Ariane Germeyer, Markus Moessner, Stephanie Bauer, Beate Ditzen, Tewes Wischmann.

**Data curation:** Maren Schick, Sabine Roesner, Ariane Germeyer, Markus Moessner, Stephanie Bauer, Tewes Wischmann.

**Formal analysis:** Franziska Kremer, Tewes Wischmann.

**Supervision:** Beate Ditzen, Tewes Wischmann.

**Writing – original draft:** Franziska Kremer.

**Writing – review & editing:** Franziska Kremer, Maren Schick, Sabine Roesner, Ariane Germeyer, Markus Moessner, Stephanie Bauer, Beate Ditzen, Tewes Wischmann.

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
