## [Decision Letter · Decision Letter 0]

7 Mar 2025

Dear Dr. Kremer,

Thank you for submitting your manuscript to PLOS ONE. After careful consideration, we feel that it has merit but does not fully meet PLOS ONE’s publication criteria as it currently stands. Therefore, we invite you to submit a revised version of the manuscript that addresses the points raised during the review process.

Based on the reviewers' feedback, major revisions are required. Particularly, add details on randomization, allocation concealment, and questionnaire language. Provide further details on statistical methods and adjust for multiple comparisons where applicable. Please remove unsupported claims from the conclusions. Finally, refine formatting and correct typographical errors.

We look forward to receiving your revised manuscript.

Kind regards,

César González-Blanch, PhD

Academic Editor

PLOS ONE

Journal Requirements:

Reviewers' comments:

Reviewer's Responses to Questions

**Comments to the Author**

1. Is the manuscript technically sound, and do the data support the conclusions?

Reviewer #1: Partly

Reviewer #2: No

2. Has the statistical analysis been performed appropriately and rigorously?

Reviewer #1: No

Reviewer #2: I Don't Know

3. Have the authors made all data underlying the findings in their manuscript fully available?

Reviewer #1: Yes

Reviewer #2: Yes

4. Is the manuscript presented in an intelligible fashion and written in standard English?

Reviewer #1: Yes

Reviewer #2: Yes

Reviewer #1: The manuscript needs English proofreading, and significant revisions are required for the results presentation.

Abstract Results: The N could be omitted. The p values for the significant results are to be included.

Line 92: The waiting period is to be clearly defined e.g what is the duration? It would be good to provide a chart or incorporate it with Figure 1 showing the days, assessment points, type of assessment etc.

Line 92-94: Days is to be used to indicate the pre intervention, post intervention and evaluation.

Line 132: For the sample size calculation using GPower, other information to be provided such as one or two-tailed, type of study design, number of groups involved.

Line 135: The person who prepared the list of names and the information on allocation concealment is to be stated.

Line 136: The name of computerised randomization system is to be provided.

Line 162: ‘also in the medical records’ is to be revised to ‘also from the medical records’

Line 164-165: Whether the language in English or German version is to be stated.

Line 168: Typo Sociographics.

Line 193: Open questions to be written as open ended questions.

Line 196-198: The sentence unclear and requires revision.

Line 198: The imputation method is to be stated.

Line 201 and Table 1: Based on the CONSORT statement, all statistical tests for group comparison at baseline are to be avoided. The randomization is supposed to distribute both known and unknown confounders evenly between the groups and the groups should be similar at baseline concerning both the measured and unmeasured variables. However, if significant differences are noted between the groups at baseline, it suggests that the randomization process may not have been successful in achieving balance between the groups.

Line 216-217: The application of ANOVA and Wilcoxon test is unclear. ANOVA is commonly used to compare means across three or more groups, but it can also be applied to two groups, although a t-test is usually preferred in such cases. For parametric data involving two related time points (e.g., pre and post), a paired t-test is the standard choice. Likewise with Line 250-252. The sentence requires revision.The specific type of Wilcoxon test is to be stated.

Line 224: Typo chi-test.

Line 226: The significance level, and multiple testing corrections where applicable are to be stated.

Line 236-237: The sentence ‘The reason for the difference ….. was determined by applying the Bonferroni correction in the married subgroup.’ requires revision. Likewise with Line 240.

Line 236: Unclear what other subgroup refers to.

Line 234-242: The findings to be indicated with table number(s)/ denoted in the table and table footnote.

Line 249: Depressivity is to be replaced with Depression.

Table 1, 2, 3, 4: Two decimal points for SD are sufficient. Likewise with confidence interval.

Table 1-8: The symbol +,‡:, §: can be omitted. The abbreviations CD, M, SD are sufficient.

Line 267-268: The statistical test is to be mentioned. The detail to be attached as supplementary table.

Line 310, 323: The p value for Bonferroni correction is to be presented. Likewise with other sections were applicable.

The presentation format and data for Table 7 and 8 requires improvements.

All the statistical tests used in the results section are to be stated in statistical analysis section.

Table 6: The statistical test used to produce this output is to be mentioned.

Line 306: chi-test to be replaced with chi-square test.

All GLM analysis results preferably to be presented in table form and attached as supplementary. The detail description and application of GLM is to be described in the statistical analysis section.

Some p value values were italicised and some were not. This needs to be standardized.

Reviewer #2: Thanks for inviting me to review this paper. The purpose of this is to reduce the depression and anxiety among couples undergoing fertility treatment.

Major revisions are needed to make it clearer. Details showed in the followings:

1. The primary objective of this study is to reduce anxiety and depression. To support this aim, it is essential to provide information on the prevalence and severity of anxiety and depression within this population.

2. It is recommended to provide a justification for including male participants, specifically addressing the relationship between anxiety and depression among couples and its relevance to the research objectives. Additionally, the results section presents data for male and female participants separately, which creates confusion regarding the purpose.

3. Additional details regarding the randomization process are needed to enhance the reproducibility of the study.

4. While the results indicate that the average scores for anxiety and depression fall below the clinical cutoff, it would be beneficial to report the percentage of participants exhibiting clinical anxiety and depression. Furthermore, discuss on the anxiety and depression scores observed in this study is suggested to inform future studies.

5. The conclusion should be firmly grounded in the statistical results obtained from the study to ensure scientific rigor and accuracy. The suggestions on using ‘face to face methods’ and ‘low mood by themselves do not impair fertility’ lack evidence support in this study.

6. Statistics methods are suggested to reviewed by a statistician.

**Do you want your identity to be public for this peer review?** For information about this choice, including consent withdrawal, please see our Privacy Policy

Reviewer #1: No

Reviewer #2: No

---

## [Author Response · Author response to Decision Letter 1]

7 May 2025

Review Comments to the Author:

Reviewer #1: The manuscript needs English proofreading, and significant revisions are required for the results presentation.

Response: Thanks for the suggestions. The text has been checked again.

Abstract Results: The N could be omitted. The p values for the significant results are to be included.

Line 92: The waiting period is to be clearly defined e.g what is the duration? It would be good to provide a chart or incorporate it with Figure 1 showing the days, assessment points, type of assessment etc.

Response: Thank you for the suggestion. We have added a new Table 1 with more detailed information and expanded Figure 1.

Line 92-94: Days is to be used to indicate the pre intervention, post intervention and evaluation.

Response: The day was noted in Figure 1.

Line 132: For the sample size calculation using GPower, other information to be provided such as one or two-tailed, type of study design, number of groups involved.

Response: Thank you for this comment. We have added the missing information (page 7, line 132-133).

Line 135: The person who prepared the list of names and the information on allocation concealment is to be stated.

Response: Thanks for pointing that out. The relevant statistician is now mentioned in line 138 of the main text.

Line 136: The name of computerised randomization system is to be provided.

Response: Thanks, it has now been mentioned in line 139.

Line 162: ‘also in the medical records’ is to be revised to ‘also from the medical records’

Response: Thanks for the hint.

Line 164-165: Whether the language in English or German version is to be stated.

Response: Thanks for the hint.

Line 168: Typo Sociographics.

Response: We have used the term sociographic variables.

Line 193: Open questions to be written as open ended questions.

Response: Thanks for the hint.

Line 196-198: The sentence unclear and requires revision.

Line 198: The imputation method is to be stated.

Response: Thank you very much, we have added the information on page 10, lines 199-203.

Line 201 and Table 1: Based on the CONSORT statement, all statistical tests for group comparison at baseline are to be avoided. The randomization is supposed to distribute both known and unknown confounders evenly between the groups and the groups should be similar at baseline concerning both the measured and unmeasured variables. However, if significant differences are noted between the groups at baseline, it suggests that the randomization process may not have been successful in achieving balance between the groups.

Response: We believe that the randomization process reflects the law of large numbers and that the group size should be slightly larger due to the only very small real intervention effects.

Line 216-217: The application of ANOVA and Wilcoxon test is unclear. ANOVA is commonly used to compare means across three or more groups, but it can also be applied to two groups, although a t-test is usually preferred in such cases. For parametric data involving two related time points (e.g., pre and post), a paired t-test is the standard choice. Likewise with Line 250-252. The sentence requires revision. The specific type of Wilcoxon test is to be stated.

Response: Both T-test and ANOVA are possible. We have decided to use ANOVA. The Wilcoxon test is the Wilcoxon test for paired samples.

Line 224: Typo chi-test.

Response: Thank you very much. We have decided on the following writing style: χ²-test.

Line 226: The significance level, and multiple testing corrections where applicable are to be stated.

Response: Thank you for pointing that out. We have set the significance level at 5%.

Line 236-237: The sentence ‘The reason for the difference ….. was determined by applying the Bonferroni correction in the married subgroup.’ requires revision. Likewise with Line 240.

Line 236: Unclear what other subgroup refers to.

Response: We have made every effort to simplify the statements and make them reader-friendly (page 12, line239-245).

Line 234-242: The findings to be indicated with table number(s)/ denoted in the table and table footnote.

Response: We have made every effort to adjust the tables accordingly.

Line 249: Depressivity is to be replaced with Depression.

Response: We have taken note of your comment and changed “Depressivity” to “Depression.”

Table 1, 2, 3, 4: Two decimal points for SD are sufficient. Likewise with confidence interval.

Response: Thank you very much. We have amended the information.

Table 1-8: The symbol +,‡:, §: can be omitted. The abbreviations CD, M, SD are sufficient.

Response: Thank you very much. We have amended the information.

Line 267-268: The statistical test is to be mentioned. The detail to be attached as supplementary table.

Response: We believe that this information is important for the body text and that it eliminates the need for an additional table.

Line 310, 323: The p value for Bonferroni correction is to be presented. Likewise with other sections were applicable.

The presentation format and data for Table 7 and 8 requires improvements.

All the statistical tests used in the results section are to be stated in statistical analysis section.

Response: In our opinion, all relevant information and statistical tests are included.

Table 6: The statistical test used to produce this output is to be mentioned.

Response: The odds ratio was calculated using SPSS and represents a standard procedure.

Line 306: chi-test to be replaced with chi-square test.

Response: As mentioned above, we use the following notation: χ²-test.

All GLM analysis results preferably to be presented in table form and attached as supplementary. The detail description and application of GLM is to be described in the statistical analysis section.

Some p value values were italicised and some were not. This needs to be standardized.

Response: Thank you very much! We have made the necessary changes.

Reviewer #2: Thanks for inviting me to review this paper. The purpose of this is to reduce the depression and anxiety among couples undergoing fertility treatment.

Major revisions are needed to make it clearer. Details showed in the followings:

1. The primary objective of this study is to reduce anxiety and depression. To support this aim, it is essential to provide information on the prevalence and severity of anxiety and depression within this population.

Response: Thank you very much, we have now included the ESHRE guideline to show the prevalence in the population.

2. It is recommended to provide a justification for including male participants, specifically addressing the relationship between anxiety and depression among couples and its relevance to the research objectives. Additionally, the results section presents data for male and female participants separately, which creates confusion regarding the purpose.

Response: Please see above. Men are affected similarly to women.

3. Additional details regarding the randomization process are needed to enhance the reproducibility of the study.

Response: Thank you very much. We have now added further information (page 8).

4. While the results indicate that the average scores for anxiety and depression fall below the clinical cutoff, it would be beneficial to report the percentage of participants exhibiting clinical anxiety and depression. Furthermore, discuss on the anxiety and depression scores observed in this study is suggested to inform future studies.

Response: Thank you very much for this important note. We have now added the missing information on pages 14 and 15 and taken it into account in the discussion.

5. The conclusion should be firmly grounded in the statistical results obtained from the study to ensure scientific rigor and accuracy. The suggestions on using ‘face to face methods’ and ‘low mood by themselves do not impair fertility’ lack evidence support in this study.

Response: Thank you very much for this improvement. We have revised lines 455 to 460 on page 23. We have improved the statement as follows: “This data is in line with research suggesting that individual daily stress levels or low mood by themselves usually do not impair fertility (Wischmann et al. 2021) and that a low-level “mini-intervention” alone therefore might not increase pregnancy rates. At the same time these results can contribute to more tailor-made face-to-face interventions that address both partners at the same time and, thereby, help couples to successfully navigate in this challenging time during ART.”

6. Statistics methods are suggested to reviewed by a statistician.

Response: Please see above.

---

## [Decision Letter · Decision Letter 1]

22 May 2025

Dear Dr. Kremer,

Thank you for submitting your manuscript to PLOS ONE. After careful consideration, we feel that it has merit but does not fully meet PLOS ONE’s publication criteria as it currently stands. Therefore, we invite you to submit a revised version of the manuscript that addresses the points raised during the review process.

We look forward to receiving your revised manuscript.

Kind regards,

César González-Blanch, PhD

Academic Editor

PLOS ONE

Journal Requirements:

Reviewers' comments:

Reviewer's Responses to Questions

**Comments to the Author**

Reviewer #1: (No Response)

2. Is the manuscript technically sound, and do the data support the conclusions?

Reviewer #1: Partly

3. Has the statistical analysis been performed appropriately and rigorously?

Reviewer #1: No

4. Have the authors made all data underlying the findings in their manuscript fully available?

Reviewer #1: Yes

5. Is the manuscript presented in an intelligible fashion and written in standard English?

Reviewer #1: Yes

Reviewer #1: Table 2: Based on CONSORT statement/requirement, the column of t/X^2 and p value and the *P ≤ .05 in the footnote are to be removed.

For paired samples, Wilcoxon signed-rank test is to be mentioned.

**Do you want your identity to be public for this peer review?** For information about this choice, including consent withdrawal, please see our Privacy Policy

Reviewer #1: No

---

## [Author Response · Author response to Decision Letter 2]

3 Oct 2025

Reviewer #1: Table 2: Based on CONSORT statement/requirement, the column of t/X^2 and p value and the *P ≤ .05 in the footnote are to be removed.

For paired samples, Wilcoxon signed-rank test is to be mentioned.

Response: Thanks for pointing that out. We removed the column of t/X^2 and p value and the *P ≤ .05 in the footnote. We have added the missing information (page 14, line 260).

---

## [Decision Letter · Decision Letter 2]

16 Oct 2025

Smartphone-Supported Positive Adjustment Coping Intervention (PACI) for Couples Undergoing Fertility Treatment: A Randomised Controlled Trial

PONE-D-24-53960R2

Dear Dr. Kremer,

We’re pleased to inform you that your manuscript has been judged scientifically suitable for publication and will be formally accepted for publication once it meets all outstanding technical requirements.

Kind regards,

César González-Blanch, PhD

Academic Editor

PLOS ONE

Additional Editor Comments (optional):

Reviewers' comments:

Reviewer's Responses to Questions

**Comments to the Author**

Reviewer #1: (No Response)

2. Is the manuscript technically sound, and do the data support the conclusions?

Reviewer #1: (No Response)

3. Has the statistical analysis been performed appropriately and rigorously?

Reviewer #1: (No Response)

4. Have the authors made all data underlying the findings in their manuscript fully available?

Reviewer #1: (No Response)

5. Is the manuscript presented in an intelligible fashion and written in standard English?

Reviewer #1: (No Response)

Reviewer #1: (No Response)

**Do you want your identity to be public for this peer review?** For information about this choice, including consent withdrawal, please see our Privacy Policy

Reviewer #1: No

---

## [Editor Report · Acceptance letter]

PONE-D-24-53960R2

PLOS ONE

Dear Dr. Wischmann,

I'm pleased to inform you that your manuscript has been deemed suitable for publication in PLOS ONE. Congratulations! Your manuscript is now being handed over to our production team.

Kind regards,

on behalf of

Dr. César González-Blanch

Academic Editor

PLOS ONE